

# Historical and future changes and present-day uncertainties of ozone in China from CMIP6 models

Shuai Li[1], Hua Zhang[2*], Qi Chen[3], Yonghang Chen[1], Qi An[4], Zhili Wang[2, 5], Xinping Wu[6]

[1] College of Environmental Science and Engineering, Donghua University, Shanghai 201620, China

[2] State Key Laboratory of Severe Weather, Chinese Academy of Meteorological Sciences, Beijing 100081, China

[3] CMA earth system modeling and prediciton center, Beijing 100081, China

[4] Meteorological Service Center for the Core Areas of the Capital, Beijing 100081, China

[5] Key Laboratory of Atmospheric Chemistry of CMA, Chinese Academy of Meteorological Sciences, Beijing 100081, China

[6] Tazhong Meteorological Station, Qiemo, Xinjiang, China

*Correspondence: Hua Zhang (huazhang@cma.gov.cn)*

**Abstract:** Ozone ($O_3$) contributes to global climate change and poses a direct threat to human health. This study analyzes historical and future changes, as well as current uncertainties, in surface $O_3$ concentrations in China, based on CMIP6 and the Tracking Air Pollution in China (TAP) dataset. The results are as follows: (1) The Multi-Model Ensemble Mean (MME) of CMIP6 simulated $O_3$ concentrations is higher during June–August (JJA), averaging 105 μg·m$^{-3}$, and lowest during December–February (DJF) at 55 μg·m$^{-3}$. (2) CMIP6 models generally underestimate $O_3$ concentrations in most regions of China, with the most significant underestimation occurring in East China. (3) The MME-simulated $O_3$ concentrations exhibit lower Bias, MAE, and RMSE over natural land surfaces compared to those over anthropogenic land surfaces. The Bias reaches its minimum under cloudy conditions and peaks under partly cloudy conditions. Furthermore, the Bias generally increases with rising PM$_{2.5}$ concentrations, however, once PM$_{2.5}$ exceeds a specific threshold, the Bias begins to decline. (4) Over the entire historical period, the MME simulates an increase of 39.3 μg·m$^{-3}$ in the annual mean surface $O_3$ concentration in China. (5) Under future SSP scenarios, MME projects generally increasing $O_3$ under weak mitigation (SSP3-7.0), with East China rising by 26.9%. Strong mitigation (SSP1-2.6) leads to widespread decreases, especially in Southwest and South China (>30 μg·m$^{-3}$). (6) Differences in climate treatment, circulation, chemistry, and precursor emissions create substantial uncertainties, emphasizing the need to understand how emissions (including precursors and PM$_{2.5}$), climate, and model processes jointly affect future $O_3$ projections.



## 1 Introduction

Ozone ($O_3$) is one of the most important trace components in the Earth's atmosphere, serving as a protective barrier for the global ecosystem and a crucial heat source in the stratosphere (Zhang et al., 2017), and its variations strongly influence the climate of the stratosphere and even the troposphere (Xie et al., 2017; Haase and Matthes, 2019; Lin and Ming, 2021), playing a critical role in controlling the temperature structure of Earth's atmosphere. Studies have shown that 90% of atmospheric $O_3$ is

concentrated in the stratosphere, with only about 10% distributed in the troposphere, however, the climatic effects caused by tropospheric $O_3$ variations can be comparable to the perturbations induced in the stratosphere (Xie and Zhang, 2014). As one of the major sources of OH radicals, $O_3$ indirectly determines the lifetime of various trace constituents in the troposphere (Levy, 1971). Additionally, $O_3$ is an important greenhouse gas, a strong oxidant, and a plant toxin, which not only influences global

climate change (Monks et al., 2015) but also directly harms human health (Shindell et al., 2012; Wang et al., 2021) and vegetation growth (Avnery et al., 2011; Lin et al., 2018; Feng et al., 2018). The Global Burden of Disease Report (GBDR) states that more than 360,000 premature deaths globally in 2019 were attributed to exposure to ambient $O_3$, and that high $O_3$ exposure may exacerbate the $PM_{2.5}$-mortality risk (Weichenthal et al., 2017). Therefore, studying the evolution of $O_3$ is of great

significance for understanding global climate change and protecting Earth's organisms.
Over the past few years, due to rapid industrial development, precursor pollutants have been continuously emitted in large quantities, causing severe $PM_{2.5}$ and $O_3$ pollution in China (Maji et al., 2018; Lu et al., 2018; Qin et al., 2021). To improve air quality, the State Council issued the "Air Pollution Prevention and Control Action Plan" (APPCAP) in 2013, with the goal of reducing $PM_{2.5}$

concentrations in the key regions of Beijing-Tianjin-Hebei, the Yangtze River Delta, and the Pearl River Delta by 25%, 20%, and 15%, respectively, by 2017. Accordingly, a series of air pollution control measures have been implemented, including optimizing industrial structure, increasing the supply of clean energy, limiting high-emission vehicles, and banning high-emission sources such as open biomass burning (Qiu et al., 2016). As a result, since the "13th Five-Year Plan", China has made

significant progress in air pollution mechanism research and control (Li et al., 2020; Lu et al., 2020; An et al., 2022; Su et al., 2022), particularly with a notable decrease in the annual average concentration of $PM_{2.5}$ in major regions.



In stark contrast to the improvements in $PM_{2.5}$ pollution control, most regions in China experienced a persistently fluctuating upward trend in annual $O_3$ concentrations during 2013–2018, with an average annual increase of 1–3 ppb (Li et al., 2019). By 2018, the national annual mean $O_3$ concentration had increased by 17.59% and 15.22% compared to 2013 and 2015 levels, respectively (Wang et al., 2020). The proportion of $O_3$-polluted days has become increasingly significant, and prolonged, large-scale $O_3$ pollution episodes have occurred more frequently, particularly in major urban agglomerations such as the Beijing-Tianjin-Hebei (BTH), Yangtze River Delta (YRD), and Pearl River Delta (PRD) regions (Dai et al., 2020; Zhao et al., 2020). Notably, since 2015 in the PRD and since 2017 in the YRD, the proportion of days with $O_3$ exceedance has surpassed that of particulate matter, making $O_3$ the primary pollutant (Lu et al., 2018; Wang et al., 2020). Despite the positive effects of policy implementation on reducing nitrogen oxides ($NO_x$) and volatile organic compounds (VOCs) in China (Lu et al., 2018), significant uncertainties remain regarding the abundance, spatial distribution, and related processes of these short-lived gases, which constrain the further optimization and effectiveness of emission control policies (Wild et al., 2020). Against this backdrop, surface $O_3$ pollution in China continues to worsen and expand. While $PM_{2.5}$ pollution has shown consistent improvement, effectively controlling $O_3$ pollution has emerged as a critical challenge for air quality management in China, posing serious difficulties for both the scientific community and policymakers. Under the guidance of the "Dual-Carbon" strategy, achieving sustained improvements in air quality through precise and science-based measures has become an urgent scientific and technological issue. Therefore, accurately understanding the spatiotemporal evolution of surface $O_3$ concentrations in China is of great importance.

Currently, ground-based observations (including surface and radiosonde measurements) (Zhan et al., 2021; Liu et al., 2022), satellite remote sensing retrievals (such as column concentrations, vertical profiles, and multi-source data fusion) (Hubert et al., 2021; Zhao et al., 2022), and model simulations (Xue et al., 2020; Morgenstern, 2021) provide essential data and analytical approaches for monitoring $O_3$ and its precursors, investigating pollution sources and transport characteristics, and evaluating the accuracy of retrieval products. Ground-based observations are known for their high accuracy, satellite remote sensing offers broad coverage, and model simulations can extend both spatially and temporally through parameterization, these methods complement each other and collectively support ozone-related research. For a long time, chemical-climate models have been essential tools for global surface $O_3$



research, capable of reproducing past and present $O_3$ distributions across different spatial and temporal

scales, and exploring their relationships with precursors and atmospheric physical-dynamic processes.

However, due to the complexity of the tropospheric $O_3$ budget mechanisms, particularly the effects of

chemical reaction chains, precursor emission distributions, and meteorological conditions, significant

discrepancies remain in model results, both among different models and compared with observations.

These discrepancies reflect the limitations of the models in parameterization of physical and chemical

processes, as well as uncertainties in emission inventories and boundary conditions. Therefore,

identifying, investigating, and quantifying the differences between models and observations is crucial

for improving and advancing model performance (Young et al., 2018).

CMIP6 and the latest IPCC AR6 adopt new emission scenarios driven by different socioeconomic

pathways, the Shared Socioeconomic Pathways (SSPs), replacing the four Representative

Concentration Pathways (RCPs) used in CMIP5, this is a significant advancement in the CMIP6

scenarios (Eyring et al., 2016; Zhou et al., 2019). Previous studies have shown that most CMIP6 Earth

System Models (ESMs) are capable of capturing the spatial distribution of global surface $O_3$

concentrations (Turnock et al., 2020; Ivanciu et al., 2021; Griffiths et al., 2021; Shang et al., 2021), but

they tend to produce an overall positive bias of 5–10% (3.6±4.4 ppbv) (Sun and Archibal, 2021), with

larger biases in the Northern Hemisphere and smaller biases in the Southern Hemisphere. This

discrepancy may be attributed to the limitations of $O_3$ precursor emission data (Young et al., 2013).

Currently, the evaluation of the latest CMIP6 simulations for surface $O_3$ in China is still limited.

Therefore, this paper based on multi-model $O_3$ products from CMIP6, conducts an analysis of the

historical and future changes of surface $O_3$ in China, as well as the associated current uncertainties.

First, using the $O_3$ dataset from the Tracking Air Pollution in China (TAP), we assess the distribution

and uncertainty of surface $O_3$ simulated by nine CMIP6 models under various conditions, including

different temperatures, cloud cover levels, complex land surface types, and pollutant concentrations,

for the period 2014–2023 across China and its seven sub-regions (Northeast China, North China, East

China, South China, Central China, Northwest China and Southwest China) (Figure 1). Secondly, the

changes in surface $O_3$ over different regions of China during the historical period 1850–2014 are

calculated to provide a background for the analysis of future changes. Then, based on different SSPs in

CMIP6 experiments, the future changes in surface $O_3$ across China are predicted and analysed for the

period 2015–2100. Finally, a comparison of different CMIP6 models under a single future scenario



(SSP3-7.0) is conducted to identify the potential causes of models differences, aiming to provide valuable references for future O3 pollution control and prediction efforts.

## 2 Research data and methods

The data used in this paper primarily includes $O_3$ products simulated by all models (9 models) in the CMIP6 chemistry models ("AERmon" CMIP6 table ID) under both historical and future scenarios (see Table 1). All data can be accessed from the World Climate Research Programme node(WCRP) (https://esgf-node.llnl.gov/search/cmip6/, last access: 8 April 2024). Specifically, all available data from 1850 to 2014 were obtained from the historical experiments of CMIP6 (Eyring et al., 2016), which studied the surface $O_3$ changes in China during the industrial period. Additionally, all available data for the period 2015–2100 from different shared economic pathways in ScenarioMIP (O'Neill et al., 2016), were used, and the specific SSP3-7.0-lowNTCF scenario from AerChemMIP (Collins et al., 2017) were used to investigate future changes in surface $O_3$ across different regions of China. To investigate the potential drivers behind the differences in future surface $O_3$ projections over China and its sub-regions under the SSP3-7.0 scenario, this study further incorporates and analyzes VOCs emission data provided by CMIP6. In CMIP6, variables representing non-methane volatile organic compound (NMVOC) emissions primarily include emivoc and emibvoc. The emivoc variable denotes the total emission rate of NMVOCs, covering both anthropogenic and biogenic sources, whereas emibvoc represents NMVOCs emissions from natural sources (e.g., vegetation), and is commonly used as a proxy for biogenic volatile organic compounds (BVOCs). For clarity and consistency, the terms NMVOCs and BVOCs are hereafter used to refer to the emission fluxes represented by emivoc and emibvoc, respectively.

For the historical experiments, CMIP6 provides $O_3$ data from 9 models and 44 ensemble members. The future scenario with the most available data is SSP3-7.0, with $O_3$ data from 9 models and 37 ensemble members. This is followed by SSP3-7.0-lowNTCF scenarios, which have $O_3$ data from 8 models and 15 ensemble members. For other scenarios from the Tier 1 experiments in CMIP6 (SSP1-2.6, SSP2-4.5), $O_3$ data from 5 models are available for analysis, while only 4 models are available for SSP5-8.5. Due to the limited availability of model data for the Tier 2 CMIP6 scenarios (SSP1-1.9,



150 SSP4-3.4, SSP4-6.0, and SSP5-3.4-over), our analysis focuses on SSP3-7.0 and the other scenarios

from the Tier 1 experiments.

**Table1. Number of ensemble members used for the historical- and future-scenario experiments from each model in the analysis of surface O₃ in this study**.

| CMIP6 Models | Institution | Historical | SSP1-2.6 | SSP2-4.5 | SSP3-7.0 | SSP3-7.0-lowNTCF | SSP5-8.5 | Model reference |
|---|---|---|---|---|---|---|---|---|
| BCC-ESM1 | Beijing Climate Center, China Meteorological Administration, China | 3 | | | 3 | 3 | | Wu et al. (2020) |
| CESM2-WACCM | National Center for Atmospheric Research, Climate and Global Dynamics Laboratory, USA | 3 | | | 3 | 3 | | Emmons et al. (2020) |
| EC-Earth3-AerChem | European Consortium of Meteorological Services, Research Institutes, and High-performance Computing Centers | 4 | | 1 | 3 | 3 | | Noije et al. (2021) |
| GFDL-ESM4 | NOAA Geophysical Fluid Dynamics Laboratory, USA | 1 | 1 | 1 | 1 | 1 | 1 | Horowitz et al. (2020) |
| ISPL-CM5A2-INCA | Institute Pierre Simon Laplace, Paris, France | 1 | | | 1 | 1 | | Sepulchre et al. (2020) |
| MIROC-ES2H | University of Tokyo, National Institute for Environmental Studies, and Japan Agency for Marine‐Earth Science and Technology, Japan | 3 | 1 | 2 | 1 | | 3 | Hajima et al. (2020) |
| MRI-ESM2-0 | Meteorological Research Institute, Japan | 10 | 4 | 10 | 5 | 3 | 5 | Yukimoto et al. (2019) |
| UKESM1-0-LL | Natural Environment Research Council, and Met Office, United Kingdom | 18 | 5 | 5 | 19 | 1 | 4 | Sellar et al. (2019) |
| UKESM1-1-LL | Natural Environment Research Council, and Met Office, United Kingdom | 1 | 3 | | 1 | | | Sellar et al. (2019) |
| **Total number of models** | | 44 | 14 | 19 | 37 | 15 | 13 | |

155

To assess the uncertainty in the CMIP6 simulation of current O₃ concentrations, this study utilizes the

TAP (http://tapdata.org.cn/, last access: 8 April 2024) O₃ monthly products for the period from January

2014 to December 2023, with a spatial resolution of 0.1°. TAP is a near-real-time atmospheric

composition tracking dataset for China, developed by Tsinghua University in collaboration with several

other institutions, which mainly includes O₃, PM₂.₅ and their major chemical components. The O₃

product is derived through a machine learning model that integrates multiple data sources, including O₃

observational data, satellite remote sensing vertical profiles of O₃, CMAQ simulations, WRF

simulations, vegetation indices, nighttime lights, and population data, to estimate daily O₃

concentrations. TAP estimates show a high correlation with in-situ observations of maximum daily

8-hour average O₃, with an R² value of up to 0.70 (Xue et al., 2020; Xiao et al., 2022). Figure 1 shows

the trends in surface pollutant concentrations across China and its sub-regions from 2000 to 2023 based

on TAP data. Overall, O₃ concentrations have shown an increasing trend (10% increase) across China

and sub-regions over the past decade, with a multi-year average concentration of 123.3 μg·m⁻³. Among





these, North China exhibits the most significant increase (20% increase), with the highest average

concentration of 141 μg·m⁻³, followed by East China with 133 μg·m⁻³, and the Northeast and Southwest

regions have lower concentrations, with average values of 114 μg·m⁻³ and 106 μg·m⁻³, respectively. In

contrast, PM$_{2.5}$ concentrations have shown a decreasing trend (35% decrease) over the past 24 years,

with a multi-year average concentration of 41 μg·m⁻³. North China again shows the most notable

decrease (42% decrease), with the highest average concentration of 57 μg·m⁻³, followed by Central

China at 54 μg·m⁻³, and the Northwest and Southwest regions have lower concentrations, with averages

of 32 μg·m⁻³ and 31 μg·m⁻³, respectively. Except for the Northwest (2015), all other regions reached

their peak concentrations in 2006–2007. The components of PM$_{2.5}$ have also shown decreasing trends,

with the most significant reductions observed in SO$_4$ and OM, while the reductions in BC and NH$_4$ are

less pronounced.


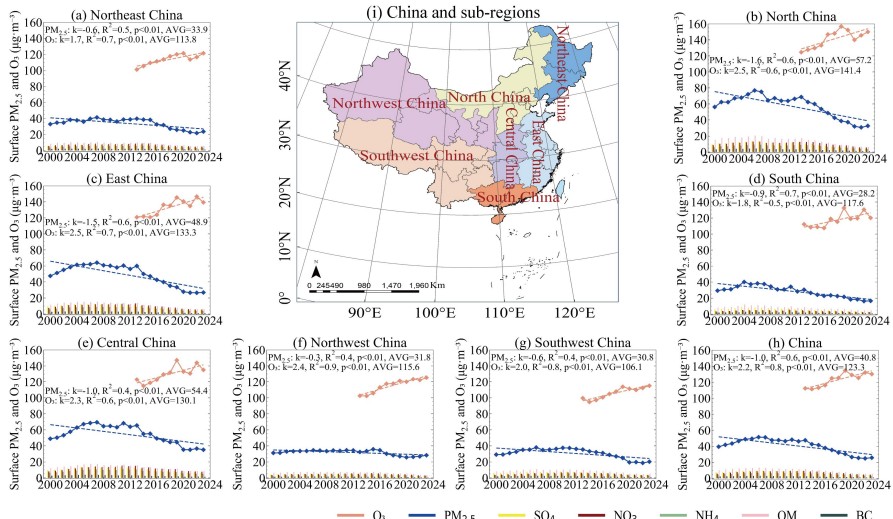

**Figure 1. An overview of the geographic location of China and seven sub-regions (Northeast China, North China, East China, South China, Central China, Northwest China and Southwest China), and the annual variations in O₃ and PM₂.₅ components from 2000–2023 based on TAP data.**


In this paper, surface O$_3$ concentration is obtained by extracting the lowest vertical layer data from the

horizontal and vertical grids of each CMIP6 model. For each model, the mean of all available ensemble

products is taken as the model's simulation results. In assessing the uncertainty of CMIP6 simulations



for current O₃ concentrations, all CMIP6 data are linearly interpolated to a resolution of 0.1° to match the TAP data.

### 3 Present-Day O₃ concentrations simulations and uncertainties

### 3.1 Under different temperature and sub-regions

Temperature directly influences O₃ production by affecting the rates of photochemical reactions and the emission of plant VOCs, such as isoprene (Coates et al., 2016). Therefore, this study compares and

analyzes the multi-year annual mean and seasonal distribution, standard deviation, and biases with TAP data for surface O₃ concentrations over China based on the Multi-Model Ensemble Mean (MME) of CMIP6 for the period 2014–2023 (Figure 2). It is evident that during summer (June, July, and August; JJA), MME shows higher O₃ concentrations, with a mean value of 105 µg·m⁻³. This is primarily due to increased photolysis activity, higher levels of oxidants, and enhanced biogenic emissions, all of which

promote O₃ formation .Additionally, high temperatures are typically associated with increased atmospheric stability and a reduction in mixing layer height, making it more difficult for O₃ to disperse and dilute, leading to its accumulation near the surface and higher concentrations(Yang et al., 2022). In spring (March, April, and May; MAM) and autumn (September, October, and November; SON), the simulated O₃ concentrations are lower than in JJA, with mean values of 89 µg·m⁻³ and 78 µg·m⁻³,

respectively. The lowest O₃ concentrations are observed in winter (December, January, and February; DJF), with a mean of 55 µg·m⁻³.

For the seven sub-regions, the seasonal patterns are generally consistent with those of the entire China (except for South China). Among them, Central China exhibits the highest O₃ concentration during JJA, reaching up to 117 µg·m⁻³, while the lowest concentration is simulated in DJF, at only 34 µg·m⁻³. In

South China, O₃ concentrations are higher in SON, with a mean value of 99 µg·m⁻³, slightly exceeding those in JJA. According to TAP data, O₃ concentrations in JJA are also lower than those in MAM and SON in South China, and only slightly higher than in DJF. This could be attributed to the stronger southeast monsoon during JJA, which drives the northward transport of O₃ and its precursors, while also bringing increased precipitation and humidity (Yin et al., 2019), thereby reducing O₃

concentrations. In Southwest China, O₃ concentrations during MAM, SON, and DJF are notably higher than in other regions, particularly in the Tibetan Plateau, where the average O₃ concentration in DJF



exceeds 90 μg·m⁻³. This may result from transboundary transport from foreign regions, especially India, where precursor emissions and $O_3$ concentrations are relatively high (Sahu et al., 2021). In Northwest China, $O_3$ concentrations remain high across all four seasons, with the average $O_3$ concentrations in

JJA reaching 113 μg·m⁻³, and in MAM and DJF, $O_3$ concentrations are second only to those in Southwest China. Northeast China, on the other hand, has the lowest $O_3$ concentrations among all sub-regions, particularly during MAM, JJA, and SON, with concentrations of only 76 μg·m⁻³, 88 μg·m⁻³, and 54 μg·m⁻³, respectively. In the experimental scenario designed by Zhang et al. (2018), which applied clean-air background concentration boundary conditions and excluded the influence of

transboundary anthropogenic emissions from foreign regions, simulated $O_3$ concentrations decreased significantly over western China but showed little change over eastern China. These results suggest that emissions from external regions primarily affect $O_3$ levels in western China, whereas domestic emissions remain the dominant contributor to $O_3$ concentrations in eastern China.

The standard deviation (SD) among the models is largest in DJF (especially in the Sichuan Basin of

Southwest China and South China), followed by JJA, smallest in SON. This suggests greater diversity in the seasonal $O_3$ cycles simulated by individual models during DJF, especially for UKESM1-0-LL and UKESM1-1-LL, which exhibit the most distinct seasonal cycles among the nine models (Figure 3), with significant negative biases in simulated $O_3$ concentrations during DJF. The spatial variation of the Bias of MME shows that $O_3$ concentrations are significantly underestimated in the eastern and northern

China, particularly during DJF. In contrast, concentrations are overestimated in Southwest China, especially in SON and JJA. The bias in Northwest China is the lowest across all seasons, with simulated $O_3$ concentrations most closely matching the TAP values.



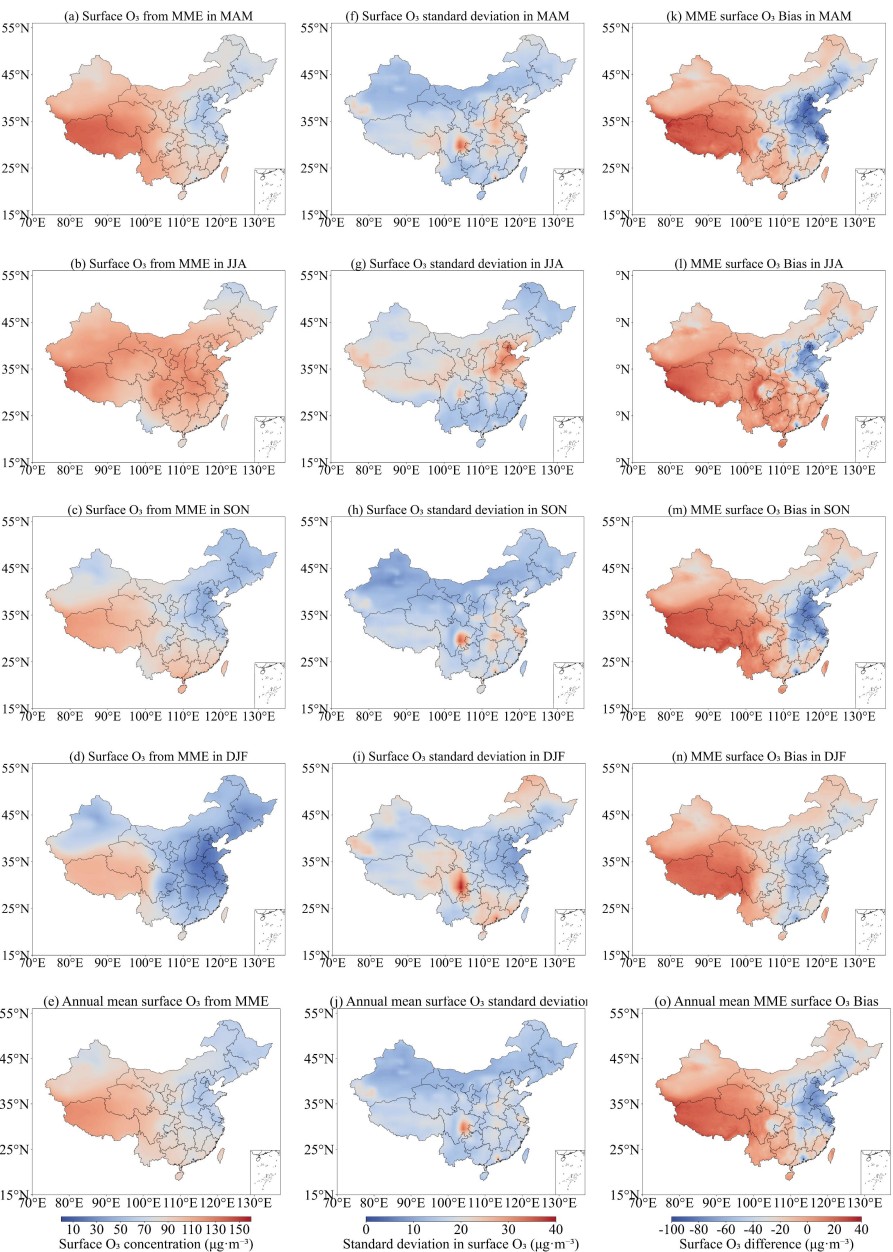

**Figure 2. Multi-model (nine CMIP6 models) annual and seasonal mean surface O₃ concentrations over the 2014–2023 period in (a) MAM; (b) JJA; (c) SON; (d) DJF; and (e) annual mean. The SD of the MME in (f)MAM, (g) DJF, (h) JJA, (i)SON, and (j) annual mean. The difference between the MME and TAP observations in (k)MAM, (l) DJF, (m) JJA, (n)SON, and (o) annual mean.**



The annual cycles of surface $O_3$ concentrations in China and its sub-regions simulated by the nine

CMIP6 models were compared with the TAP-derived values (Figure 3), it can be seen that the

correlation between the two is generally good in most regions (r > 0.73), which suggests that the

seasonality of the circulation patterns, stratosphere-troposphere exchange, and natural emissions are

well captured. However, (1) the timing of $O_3$ peak concentrations in the CMIP6 models (mostly in

July–August) is slightly delayed compared to TAP (mostly in May–June), which is consistent with the

results of the ACCMIP models (Young et al., 2018). (2) The nine CMIP6 models evaluated in this

study exhibit significant underestimation of $O_3$ concentrations across most sub-regions of China, with

the most severe underestimations found in UKESM1-0-LL and UKESM1-1-LL, except for slight

overestimations in Southwest China from May to September, all other regions show underestimations,

particularly in East and Central China during DJF, where the simulated $O_3$ concentrations are nearly 60

$\mu g \cdot m^{-3}$ lower than those calculated by TAP. This is consistent with the findings of Turnock et al.

(2020), and may be due to excessive $NO_x$ titration in the UKESM1-0-LL model, leading to an

underestimation of surface $O_3$ concentrations over much of the Northern Hemisphere's continental

regions during DJF. (3) In contrast, a small number of CMIP6 models evaluated in this study, including

BCC-ESM1, EC-Earth3-AerChem, and MRI-ESM2-0, show a certain degree of overestimation in

surface $O_3$ concentrations throughout the year in Southwest China (with an average overestimation of

30 $\mu g \cdot m^{-3}$), Northwest China (with an average overestimation of 10 $\mu g \cdot m^{-3}$), and during JJA in South

China. This may result from common sources of error in the models, such as uncertainties in emission

inventories, deposition processes, or vertical mixing (Wild et al., 2020). Additionally, the coarse

resolution of ESMs may lead to overestimation of $O_3$ concentrations in polluted areas, while

265   higher-resolution models and better consistency between nested models may improve the accuracy of

simulated surface $O_3$ concentrations (Neal et al., 2017).





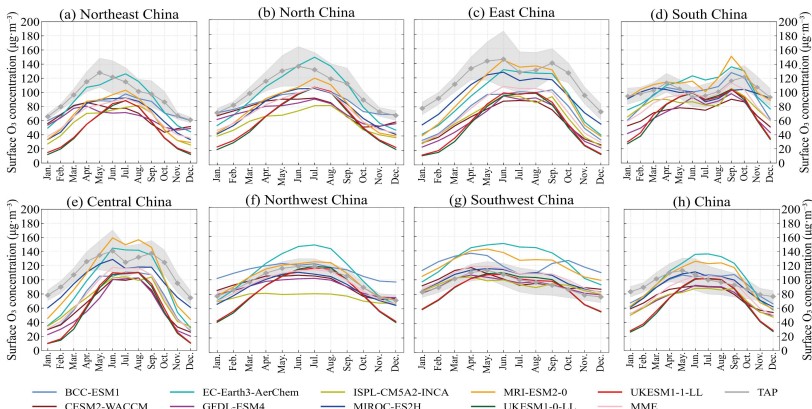

**Figure 3. Comparison of the annual cycle of O₃ concentrations, between individual CMIP6 models, the MME and TAP in China and sub-regions for the period 2014–2023. The shading shows SD of TAP observations within the region.**

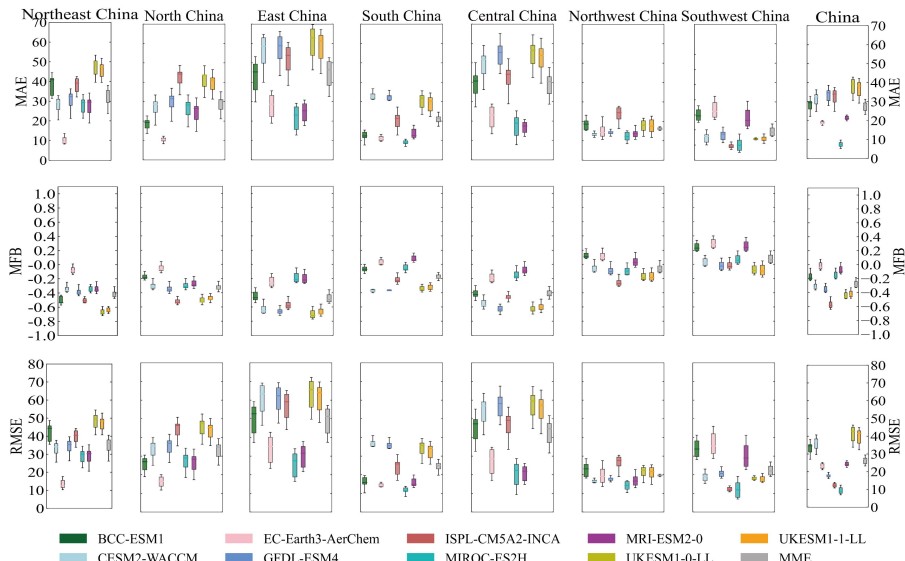

**Figure 4. Distribution of differences for O₃ concentrations (µg·m⁻³) from nine CMIP6 models in China and sub-regions during 2014–2023. The box plots show the 25th and 75th percentiles as solid boxes, median values as solid lines, dots represent the concentrations from MAE, MFB and RMSE, and whiskers extending to the minimum and maximum.**



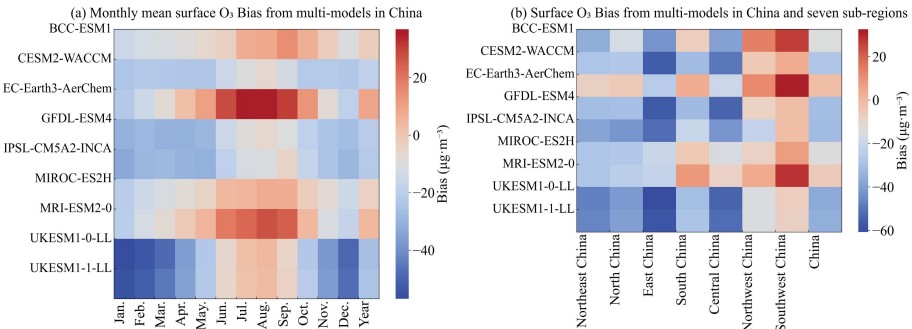

**Figure 5. Heatmap of O₃ concentrations Bias from nine CMIP6 models compared to TAP across different**
**months and regions in China and sub-regions.**

Figure 4 presents a comparison of the Mean Absolute Error (MAE), Mean Fractional Bias (MFB), and

Root Mean Square Error (RMSE) between the CMIP6 models and TAP for surface O₃ concentrations.

Combined with the Bias for different months and sub-regions (Figure 5), correlation coefficients, and

SD (Figure 6), it can be observed that the MME simulates the O₃ concentration for China with the

smallest Bias in June and the largest in January. Among the sub-regions, the simulation results for

Northwest China are the most accurate, with high correlation, and the smallest MAE, MFB, RMSE, SD,

and Bias. In contrast, the largest MAE, MFB, RMSE, SD, and Bias are found in East China,

particularly in the autumn and winter. For individual models, EC-Earth3-AerChem shows the smallest

annual average Bias for O₃ concentrations in China, with an MFB close to zero; BCC-ESM1 exhibits

the best correlation; MIROC-ES2H has the smallest MAE and RMSE, providing relatively good

simulation results; while UKESM1-0-LL has the largest MAE, MFB, RMSE, SD, and Bias. Among the

sub-regions, MIROC-ES2H provides relatively good simulations of O₃ concentrations in South China,

while UKESM1-0-LL shows the largest MAE, MFB, RMSE, SD, and Bias in East China. Thus,Overall,

the MME of CMIP6 performs better in simulating O₃ concentrations during JJA, with larger

discrepancies observed in DJF. The simulation in Northwest China is closest to TAP, while the largest

discrepancies occur in East China. EC-Earth3-AerChem is better suited for simulating or forecasting

the annual average O₃ concentrations over China, while MIROC-ES2H is more appropriate for

error-sensitive scenario analyses, BCC-ESM1 demonstrates superior performance in terms of

correlation and temporal consistency, whereas both UKESM1-0-LL and UKESM1-1-LL exhibit higher

simulation uncertainties.



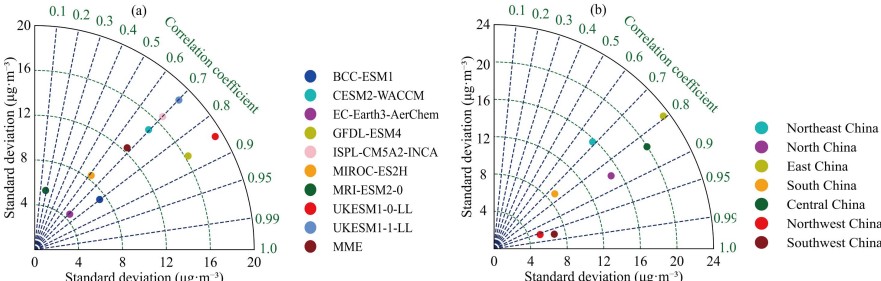

**Figure 6. Taylor diagram of the annual mean surface O$_3$ concentrations simulated by nine CMIP6 models**
**compared with the TAP data during 2014–2023 in China and sub-regions. The radial coordinate shows the**
**standard deviation in the spatial pattern, normalized by the observed standard deviation. The azimuthal**
**variable shows the correlation of the modeled spatial pattern with the observed spatial pattern.**

## 3.2 Under different underlying surface types

Vegetation type, land cover, and land use changes can influence biogenic emissions, which in turn
affect the accuracy of model-simulated O$_3$ concentrations (Ashworth et al., 2012). To investigate the
impact of these factors on simulation discrepancies and minimize the interference of temperature
changes, this study focuses on the JJA period in Northwest China, which has similar climatic
conditions and complex surface types. The selected typical underlying surfaces include natural land
surfaces such as grassland, forest, desert, and snow and ice (perennial snow), as well as anthropogenic
land surfaces such as cropland and urban. By comparing the Bias, MAE, RMSE, and SD of
MME-simulated surface O$_3$ concentrations relative to TAP (Figure 7), it is observed that TAP
simulates lower O$_3$ concentrations over natural land surfaces compared to anthropogenic land surfaces,
and the MME simulations generally follow this trend. However, the MME simulation results show the
highest O$_3$ concentrations over snow and ice surfaces and the lowest over cropland surfaces. Overall,
the MME simulations exhibit lower Bias, MAE, and RMSE for natural land surfaces compared to
anthropogenic land surfaces, with the best performance over forest and desert surfaces, and the worst
performance over urban surfaces, followed by cropland and snow and ice surfaces.

This is likely mainly due to the fact that natural land surfaces have relatively consistent physical and
chemical properties, with less human influence, resulting in more accurate O$_3$ concentration
simulations. In contrast, urban surfaces, due to intense human activities and diverse pollution sources
(such as transportation and industrial emissions), present a more complex environment, making the



processes of $O_3$ formation and destruction more intricate and leading to larger discrepancies in the simulation results.

The high albedo of snow and ice surfaces significantly reduces the absorption of solar radiation by the surface, thereby leading to a decrease the intensity of photochemical reactions, particularly the rate of $NO_2$ photolysis that leads to $O_3$ formation. In addition, the snow and ice surfaces have a weaker capacity to adsorb $O_3$, with a deposition velocity (0.03 cm·s$^{-1}$) typically lower than other natural    land surfaces such as vegetation or soil (Wesely et al., 1981). At the same time, the amount and composition

of deposited trace gases, solar irradiance, snow temperature, and the underlying materials beneath the snowpack (e.g., glacier ice, sea ice, frozen soil, and "warm" mid-latitude soils) also affect the process control, intensity, and direction of $O_3$ flux (Helmig et al., 2007). However, current atmospheric chemistry models may not adequately account for these specificities when simulating $O_3$ deposition on snow and ice surfaces. For example, deposition rate parameterisation schemes are often based on

observations of underlying surface such as vegetation and soil, which do not accurately reflect the physical and chemical properties of snow and ice surfaces. Therefore, compared to other natural land surfaces, the simulation errors in $O_3$ concentrations over snow and ice surfaces are larger, especially in high-latitude regions or areas with significant snow and ice cover during winter. This also highlights the importance of developing more detailed snow-$O_3$ exchange parameterizations for improving

models.

The simulation bias over cropland may stem from the fact that croplands are often associated with agricultural activities (e.g., fertilization and irrigation), which release large amounts of $NO_x$ and VOCs, thereby increasing the complexity of $O_3$ formation. Furthermore, changes in vegetation types and management practices in agricultural land can also influence biogenic emissions, further affecting the

simulation of $O_3$ concentrations and leading to larger model biases.



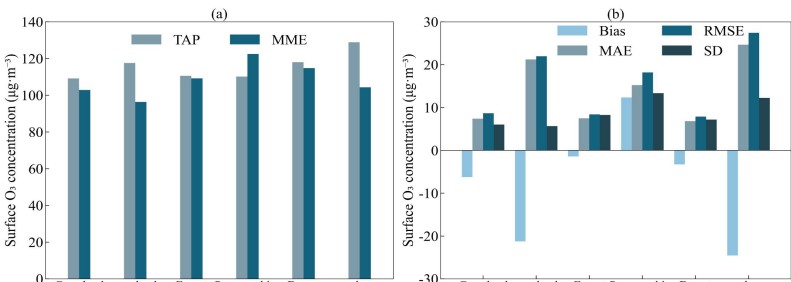

**Figure 7. The bias, MAE, RMSE and SD of surface O$_3$ concentrations simulated by CMIP6 models relative to TAP from 2014 to 2023 over the complex underlying surface in China.**


### 3.3 Under different level of total cloud cover

Referring to the definition of cloud cover levels in weather forecast and the classification of cloud cover level by Han and Cong (2015), the total cloud cover ranging from 0 %–10 % is defined as clear sky, 20 %–30 % as partly cloudy, 40 %–70 % as cloudy, and 80 %–100 % as overcast. This study

analyzes the Bias of surface O$_3$ concentrations simulated by MME relative to those simulated by TAP under different total cloud cover levels (Fig. 8). Analysis results show that the TAP simulation exhibits the highest surface O$_3$ concentrations under partly cloudy conditions, while under other cloud cover categories, O$_3$ concentrations generally decrease with increasing total cloud cover (except during JJA). This is primarily attributed to the attenuation of solar radiation by clouds and associated precipitation

processes. Clouds reduce incoming solar radiation, thereby slowing photochemical O$_3$ production. In addition, wet deposition removes certain precursors, further suppressing O$_3$ formation. Under partly cloudy conditions, however, the atmosphere is generally more stable with weaker vertical mixing, allowing O$_3$ to accumulate near the surface. However, during JJA, the relationship becomes more complex, high pollutant loads and intricate meteorological conditions likely counteract the influence of

total cloud cover, leading to a less straightforward association between cloud amount and O$_3$ concentrations.

In contrast, the MME simulations do not fully reproduce this pattern. The annual mean bias is smallest under cloudy conditions and largest under partly cloudy conditions. On a seasonal scale, the smallest bias occurs under clear-sky conditions during JJA, while the largest bias is found under partly cloudy

conditions. These discrepancies may stem from the complex interactions through which cloud cover influences O$_3$ by modulating shortwave radiation, photochemical rates, and meteorological variables





such as temperature, precipitation, and boundary layer height. They are also closely tied to structural differences among models in physical parameterizations, radiative transfer schemes, and chemical mechanisms.

Therefore, when using CMIP6 models for $O_3$-related assessment and projection, it is essential to adequately account for the interactions among cloud cover, precipitation, and other key meteorological factors, particularly under polluted and complex meteorological conditions, in order to reduce model uncertainties and improve simulation accuracy (Jacob and Winner, 2009).

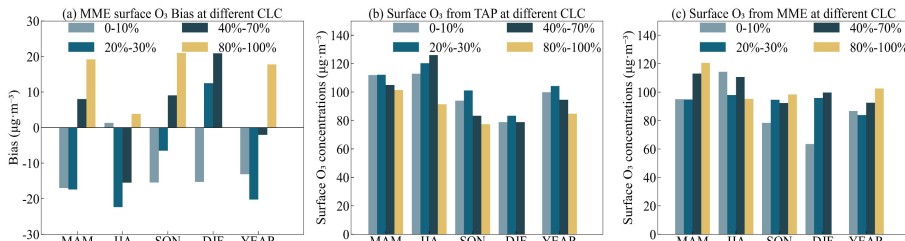


**Figure 8. The bias in surface $O_3$ concentrations simulated by CMIP6 models relative to TAP from 2014 to 2023 under different total cloud cover levels in China**.

### 3.4 Under different concentrations of PM$_{2.5}$ and its components

Aerosols play a crucial role in the simulation of $O_3$ concentrations, Variations in PM$_{2.5}$ concentrations can influence $O_3$ concentrations by altering the chemical composition and light absorption characteristics of aerosols, as well as the impact on solar radiation, which in turn affect the rate of photochemical reactions. Lou et al. (2014) used GEOS-Chem simulations to show that the mean Bias in $O_3$ concentrations for the China is 9% when aerosols are considered, compared to 33% when

aerosols are not considered. Therefore, this study analyzes the Bias in surface $O_3$ concentrations simulated by the MME relative to TAP under different PM$_{2.5}$ levels(Figure 9). The results show that during the JJA and SON, when PM$_{2.5}$ concentrations are relatively low, TAP data indicate an increase in $O_3$ concentrations with rising PM$_{2.5}$ levels. However, during the DJF, when PM$_{2.5}$ concentrations are higher, $O_3$ concentrations decrease as PM$_{2.5}$ levels increase. This is primarily due to the fact that,

during JJA and SON, although the increase in PM$_{2.5}$ concentrations may have some localized inhibitory effects on $O_3$ formation, the abundant sunlight and favorable meteorological conditions promote $O_3$





generation. In contrast, during DJF, due to insufficient sunlight, stronger atmospheric stability, and

higher $NO_x$ concentrations, $NO_x$ titration is more likely to occur, which suppresses $O_3$ formation. At the

same time, the increase in BC concentration in $PM_{2.5}$ (Figure S1) enhances the light absorption of $PM_{2.5}$,

further reducing UV radiation intensity and thus inhibiting $O_3$ photochemical production. Additionally,

other components of $PM_{2.5}$, such as $NO_3$ and OM, may also affect $O_3$ concentrations through various

pathways. These factors collectively lead to a decrease in $O_3$ concentration when $PM_{2.5}$ increases.

The MME simulation results indicate that surface $O_3$ concentrations generally decrease with increasing

$PM_{2.5}$ levels (except during JJA). However, under extreme pollution conditions in DJF, $O_3$

concentrations exhibit a slight increase once $PM_{2.5}$ exceeds 125 $\mu g \cdot m^{-3}$. This suggests a complex

nonlinear relationship between $PM_{2.5}$ and $O_3$ formation. Previous studies have shown that reducing

aerosol emissions without corresponding cuts in precursor pollutants could lead to increased surface $O_3$

over eastern China (Li et al., 2018), indicating that aerosols suppress $O_3$ production through light

attenuation and heterogeneous reactions. At low to moderate $PM_{2.5}$ concentrations, the increase in

$PM_{2.5}$ largely inhibits $O_3$ formation by scavenging key radicals (e.g., $HO_2$ and $NO_x$) and reducing solar

radiation intensity, thereby leading to a decline in $O_3$. However, under extreme pollution conditions,

especially in winter, this suppressing effect tends to saturate. Meanwhile, the nocturnal titration of $O_3$ is

weakened under high $NO_x$ conditions, which, combined with unfavorable meteorological conditions

such as temperature inversions and a lower boundary layer, results in $O_3$ concentrations no longer

decreasing with further increases in $PM_{2.5}$ and may even slightly increase.

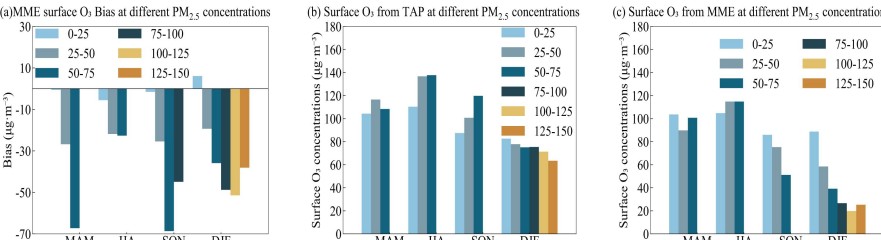

**Figure 9. The bias in surface $O_3$ concentrations simulated by CMIP6 models relative to TAP from 2014 to 2023, under different $PM_{2.5}$ concentrations in China.**


These findings highlight that the synergistic and inhibitory effects between $PM_{2.5}$ and $O_3$ vary

significantly across seasons and pollution levels. This implies that future air pollution control strategies



should adopt coordinated mitigation of both $PM_{2.5}$ and $O_3$ precursors to avoid potential side effects

from single-pollutant reduction policies.

**4 $O_3$ from the pre-industrial period to present day**

This study analyzes the annual mean surface $O_3$ concentration changes in China and its sub-regions

from pre-industrial to present times based on 9 CMIP6 models and MME relative to the 2014–2023

mean (Figure 10). The MME results show that, since 1850, the annual mean surface $O_3$ concentration

in China has increased by 39.3±14.4 $\mu g \cdot m^{-3}$ (±1 SD), with the maximum change of 57.9 $\mu g \cdot m^{-3}$ (from

the MIROC-ES2H model) and the minimum change of 23.1 $\mu g \cdot m^{-3}$ (from the UKESM1-0-LL model).

Before 1950, the annual mean increase in $O_3$ concentration was relatively slow, at only 0.12 $\mu g \cdot m^{-3}$;

however, after 1950, the rate of increase accelerated significantly, with an annual mean increase of 0.28

$\mu g \cdot m^{-3}$. This change is likely primarily related to the significant increase in anthropogenic activities

during this period, especially the substantial increase in anthropogenic precursor emissions, such as

$CH_4$, $NO_x$, CO, and NMVOCs. The simulations of historical $O_3$ concentrations by different CMIP6

models show that the EC-Earth3-AerChem model yields the highest values, while the

IPSL-CM5A2-INCA model yields the lowest. These differences reflect variations in aerosol, climate,

and atmospheric chemistry process simulations across different models. Griffiths et al. (2021), based

on ground, sounding, and satellite data from the past few decades, assess the performance of multiple

CMIP6 models in simulating surface $O_3$ concentrations. Their study indicates that these models are

capable of accurately reproducing the spatial distribution, seasonal variation, and interannual variability

and trends of surface $O_3$ concentrations. This also indicates the reliability of CMIP6 models in

simulating historical surface $O_3$ variations, which provides support for their future projections of $O_3$

concentrations under climate change scenarios.




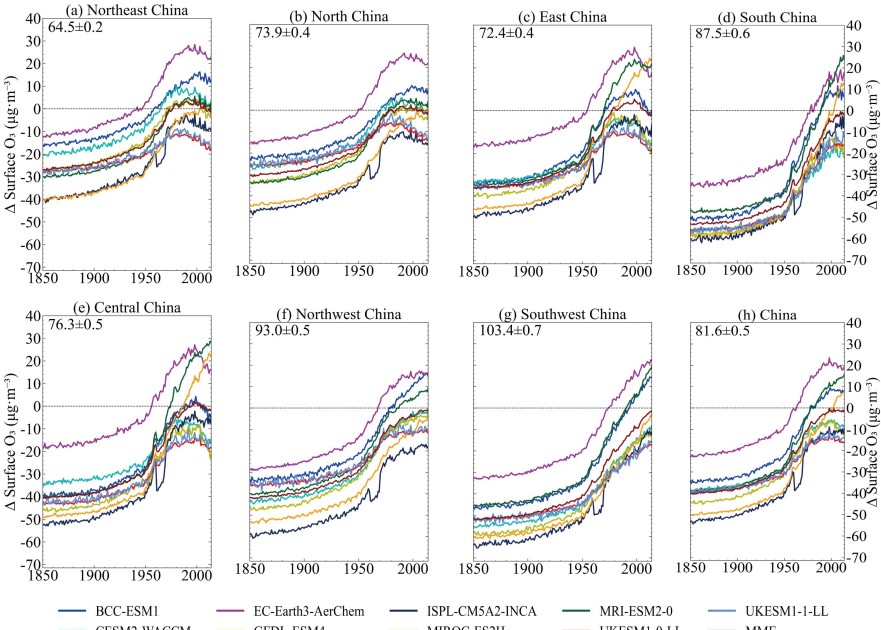

**Figure 10. Changes in the China and sub-regions annual mean surface O₃ concentrations from the pre-industrial period to present day, relative to a 2014–2023 mean value, across nine CMIP6 models and MME. The Multi-model annual mean 2014 – 2023 surface O₃ concentrations (±1 SD) are shown in the top left of each panel.**

The historical changes in surface O₃ concentrations simulated by different CMIP6 models show significant regional variations, the UKESM1-0-LL model tends to simulate the smallest historical O₃ changes (with the smallest change in Northeast China); while the MIROC-ES2H model simulates the largest O₃ changes (with the largest change in Central China), followed by MRI-ESM2-0 and BCC-ESM1. Although the UKESM1-0-LL simulation has a smallest historical surface O₃ response, it exhibits a larger tropospheric O₃ change during the historical period compared to other CMIP6 models (Griffiths et al., 2021). Moreover, the O₃ change simulated by UKESM1-0-LL is similar to the changes driven solely by precursor emission variations (Turnock et al., 2020), suggesting that this model may be highly sensitive to changes in emission sources when simulating O₃ responses. Central China exhibits the largest discrepancies in historical surface O₃ changes among the CMIP6 models, with the maximum difference reaching 29.6 µg·m⁻³. Turnock et al. (2020) suggest that the large differences in surface O₃ responses across CMIP6 models can be attributed to variations in the magnitude of simulated O₃ concentrations during the 1850s and the regional average O₃ concentration change rates,





which are closely related to the differing chemical sensitivities of $O_3$ formation processes to $NO_x$

concentration changes across models. Additionally, the significant changes in $PM_{2.5}$ concentrations in

this region (Su et al., 2022) may affect the $O_3$ formation process by altering the heterogeneous loss rate

of aerosols by radicals. Future simulations should further consider the impact of aerosols on $O_3$ to

improve the accuracy of surface $O_3$ concentration simulations.

**5 $O_3$ from present day to 2100**

Figure 11 shows the future changes in surface $O_3$ concentrations across China and its sub-regions under

different CMIP6 scenarios (relative to the 2014–2023 mean). Overall, it is projected that by 2100,

surface $O_3$ concentrations in China will decrease in most scenarios, with sub-regions responding

similarly to the national trend, though with varying magnitudes (Figure 12). In the Tier 1 experiment,

under the SSP1-2.6 scenario, which involves low radiative forcing, strong climate mitigation, and

significant air pollution reduction, surface $O_3$ concentrations in China are projected to decrease by

$12.6\pm3.1$ $\mu g \cdot m^{-3}$ ($\pm1$ SD of the MME) by 2050 relative to the 2013–2024 annual mean, and decrease by

$25.3\pm7.2$ $\mu g \cdot m^{-3}$ by 2100, with a reduction of approximately 32%. Due to substantial reductions in

precursor emissions, a decrease in $CH_4$ concentrations, and relatively small climate changes, under this

scenario, surface $O_3$ concentrations in all sub-regions also show significant declines. Projections

indicate that by 2100, surface O3 concentrations in the Southwest and South China regions will

decrease by more than 30 $\mu g \cdot m^{-3}$, while in East China, which experiences the smallest reduction, $O_3$

concentrations will still decrease by nearly 20 $\mu g \cdot m^{-3}$.

For the medium forcing SSP2-4.5 scenario, it is projected that by 2100, the annual mean surface $O_3$

concentration in China will decrease by $13.6\pm7.2$ $\mu g \cdot m^{-3}$, with a reduction of 17%. Meanwhile in this

scenario, the projections show that the annual mean surface $O_3$ concentrations in all sub-regions of

China will slightly increase in 2055 compared to the 2014–2025 mean, and then start to decrease, by

2100, the reduction will exceed 10 $\mu g \cdot m^{-3}$, with the most significant decrease occurring in South China,

where it may reach 24.5 $\mu g \cdot m^{-3}$. This change is primarily driven by enhanced control of precursor

emissions, relatively small climate changes, and variations in $CH_4$ concentrations.

In the SSP3-7.0 scenario, due to weak climate mitigation and weak air pollutant reduction, the annual

mean surface $O_3$ concentration in China is projected to increase by $8.4\pm2.0$ $\mu g \cdot m^{-3}$ by 2050, and





increase by 13.9±4.0 μg·m$^{-3}$ by 2100, with an increase of 17%. In this scenario, the annual mean

surface O$_3$ concentrations in all seven sub-regions show an upward trend, with the largest increase in

East China, where the concentration is expected to rise by 19.3±6.9 μg·m$^{-3}$ by 2100, with an increase of

27%. Although emissions of O$_3$ precursors such as NO$_x$ are projected to start decreasing around 2040

(Figure S3), the surface O$_3$ concentrations in all sub-regions continue to increase, indicating the

importance of changes in chemical composition, increasing CH$_4$ concentrations, and climate change in

the simulation of surface O$_3$ under the SSP3-7.0 scenario (Turnock et al., 2020; Young et al., 2013; Li

et al., 2019). Additionally, the projected differences among CMIP6 models are most pronounced in

Central and East China, suggesting some divergence in the model simulations of O$_3$ in these regions.

In the SSP3-7.0-lowNTCF scenario (Tier 2 experiment), strong carbon emission control measures are

implemented on top of the weak climate mitigation of the SSP3-7.0 scenario, along with a substantial

reduction in short-lived climate forcers (SLCFs), including BC and O$_3$ precursors, these measures

significantly improve air quality and slow down climate change. Consequently, the projections show

that under the SSP3-7.0-lowNTCF scenario, the increase in surface O$_3$ concentrations in China is

slower than in the SSP3-7.0 scenario, with relatively lower concentrations. By 2050, the annual mean

surface O$_3$ concentration in China is projected to increase by only 5.8±1.5 μg·m$^{-3}$, and by 2100, it will

increase by 4.9±2.0 μg·m$^{-3}$, representing a 6% increase. In this scenario, by 2100, surface O$_3$

concentrations in China and most of its sub-regions are expected to return to or be close to the

2014–2023 levels (especially in the Northwest China), showing a significant improvement in surface

O$_3$ pollution compared to the SSP3-7.0 scenario. However, compared to other regions of the world, the

additional reduction in precursor emissions under the SSP3-7.0-lowNTCF scenario has a relatively

small impact on improving surface O$_3$ pollution in China. This is mainly due to the increase in surface

O$_3$ concentrations in eastern China (especially in Central and Eastern China). This increase is caused

by a slight rise in NMVOCs emissions and a reduction in O$_3$ titration due to a significant decrease in

NO$_x$ emissions (Turnock et al., 2020). Additionally, the decrease in PM$_{2.5}$ concentrations under the

SSP3-7.0-lowNTCF scenario leads to a reduction in the heterogeneous loss of free radicals, which may

also contribute to the rise in surface O$_3$ concentrations (Li et al., 2019).

In the SSP5-8.5 scenario, characterized by high radiative forcing, weak climate mitigation, and weak

air pollutant emission reductions, the annual mean surface O$_3$ concentration in China is projected to

increase by 6.3±1.6 μg·m$^{-3}$ by 2050. However, by 2100, the surface O$_3$ concentration is expected to



decrease by 3.4±2.9 μg·m$^{-3}$ relative to the 2013–2024 mean, a reduction of approximately 4.2%. The

projected changes in surface $O_3$ concentrations for the sub-regions in this scenario are similar to those

in the SSP3-7.0-lowNTCF scenario (with a correlation of up to 0.7), likely due to comparable levels of

air pollutant emissions and climate change. By 2050, the surface $O_3$ concentration in most sub-regions

will increase slightly faster in the SSP5-8.5 scenario than in the SSP3-7.0-lowNTCF scenario (except

for Northeast and North China), but slower than in the SSP3-7.0 scenario (except for Central and East

China). This may be attributed to the different changes in $CH_4$ emissions under different scenarios.

Additionally, more CMIP6 model data are available for the SSP3-7.0 scenario (9 models) compared to

the SSP5-8.5 scenario (4 models), which may also influence the MME response.

In the Tier 2 experiment, under the SSP1-1.9 scenario, the annual mean surface $O_3$ concentration in

China is projected to decrease by 16.6±7.1 μg·m$^{-3}$ by 2050, and decrease by 25.3±9.5 μg·m$^{-3}$ by 2100,

with a reduction of approximately 32%. Although the SSP1-1.9 scenario represents weak climate

mitigation and weak air pollutant emission reductions, the simulated results indicate a significant

decrease in surface $O_3$ concentrations. This phenomenon may be closely related to the reduction of

$PM_{2.5}$ emissions in China and the complex effects of climate change. Specifically, climate change not

only alters the rates of chemical reactions in the atmosphere but also impacts convection activities and

the distribution of pollutants, thereby inhibiting $O_3$ formation or altering the balance between its

formation and consumption. Furthermore, the reduction of $PM_{2.5}$ may further exacerbate the decline in

surface $O_3$ concentrations by influencing atmospheric photochemical processes or altering the

concentrations of $O_3$ precursors.

Under the SSP4-3.4 and SSP4-6.0 scenarios, which represent moderate climate mitigation with

moderate air pollutant emission reductions, and under the SSP5-3.4-over scenarios which represents

moderate climate mitigation with stronger air pollutant emission reductions, the surface $O_3$

concentration in China is projected to increase by 13.5±1.3 μg·m$^{-3}$, 18.3±1.4 μg·m$^{-3}$, and 12.6±1.6

μg·m$^{-3}$ by 2050, respectively. By 2100, the surface $O_3$ concentrations are expected to decrease by

9.2±8.3 μg·m$^{-3}$, 1.6±5.9 μg·m$^{-3}$, and 13.9±11.7 μg·m$^{-3}$ respectively. This trend indicates that strong air

pollutant emission reduction measures will play a significant role in controlling future $O_3$

concentrations, improving air quality and mitigating negative climate impacts.

In summary, projections from the CMIP6 models suggest that mitigating surface $O_3$ pollution across

China will require not only reducing greenhouse gas emissions to moderate future climate change but





also implementing enhanced controls on emissions of $O_3$ precursors (including $CH_4$). Under scenarios

with pronounced climate change impacts, such as SSP3-7.0 and SSP5-8.5, the stringency of controls on

key $O_3$ precursors, particularly NMVOCs and $NO_x$, beyond 2050 is expected to lead to divergent

regional responses in long-term surface $O_3$ trends.

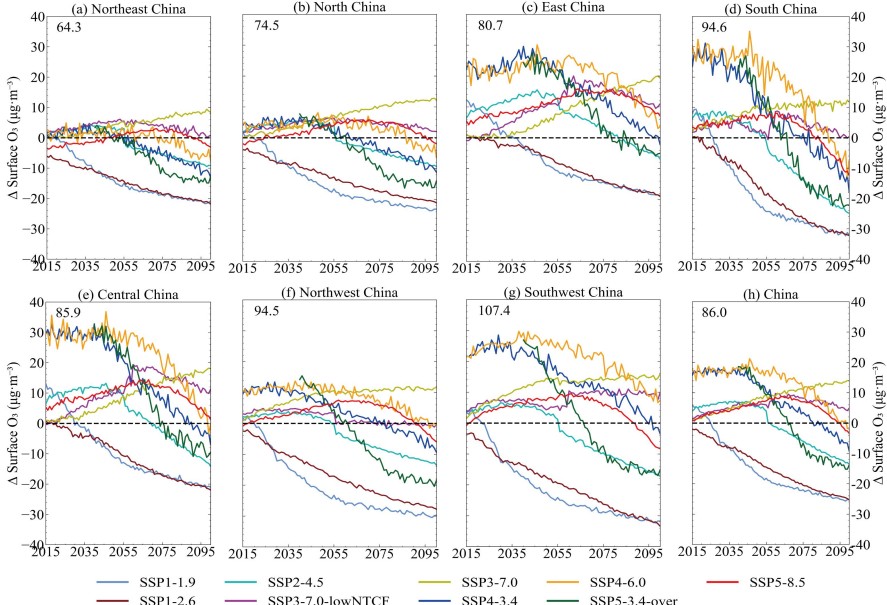

**Figure 11. Future China and sub-regions changes in annual mean surface $O_3$ for the different SSPs used in**
**CMIP6. The dashed black line represents the curve of the difference at zero. The multi-model regional mean**
**value for the years 2014–2023 mean value is shown in the top left corner of each panel.**

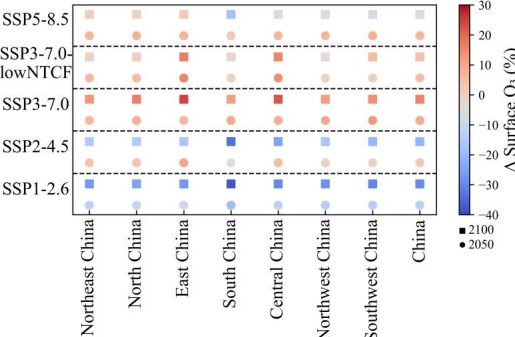

**Figure 12. Percentage change in 2050 (circles) and 2100 (squares), relative to 2015, for annual mean of $O_3$**
**across China and sub-regions in the four Tier 1 future CMIP6 scenarios and the SSP3-7.0-lowNTCF**
**scenario.**



Since the number of available CMIP6 models is the greatest under the SSP3-7.0 scenario (Table 1), this paper conducts a comparative analysis of the changes in surface $O_3$ concentrations across China and its sub-regions under the SSP3-7.0 scenario, aiming to identify the potential causes of model discrepancies.

Figure 13 illustrates the changes in the annual mean and seasonal mean surface $O_3$ concentrations in China and its sub-regions for the years 2050 (2045–2055 mean) and 2095 (2090–2100 mean) relative to the 2014–2023 mean baseline, based on different CMIP6 models under the SSP3-7.0 scenario. $O_3$ is not directly emitted into the troposphere but is produced through photochemical oxidation of CO, $CH_4$, and NMVOCs in the presence of NO and $NO_2$. The abundance of tropospheric $O_3$ is determined by its

budget, which includes chemical production, stratospheric transport, chemical loss, and deposition to the surface (Lelieveld and Dentener, 2000), and the intensity of these processes is highly sensitive to current climate conditions and the emissions and distribution of $O_3$ precursors (including $NO_x$, NMVOCs, CH4, etc.). Therefore, this study further analyzes the correlation between future annual mean surface $O_3$ concentrations under the SSP3-7.0 scenario and other variables, including $CH_4$

concentrations, Near-Surface Air Temperature (TAS), $NO_x$ concentrations, total emissions of NMVOCs and BVOCs (Figure 14).

It can be observed that surface $O_3$ concentrations predicted by different CMIP6 models under the SSP3-7.0 scenario exhibit significant regional discrepancies. In particular, in Central China, the $O_3$ concentrations predicted by MRI-ESM2-0 and EC-Earth3-AerChem are nearly twice as high as those

predicted by UKESM1-0-LL and GFDL-ESM4 (Figure S2). The lower annual mean $O_3$ concentrations in Central China for UKESM1-0-LL and GFDL-ESM4 are primarily attributed to higher $NO_x$ emissions under the SSP3-7.0 scenario. In this region, $NO_x$ emissions are approximately 2–3 times higher than those in the other two models (Figure S2), which likely triggers $NO_x$ titration and results in lower simulated surface $O_3$ concentrations. In contrast, in the MRI-ESM2-0 and EC-Earth3-AerChem,

$NO_x$ titration is rare during DJF, and the $CH_4$ concentration is higher in the EC-Earth3-AerChem model (Fig. S4), resulting in higher simulated $O_3$ concentrations in Central China. These discrepancies highlight that, although the driving factors related to $O_3$ changes (such as climate change and pollutant emissions) are crucial in all models (Figure 14), the differences in precursor emissions ($NO_x$ and $CH_4$) and chemical process responses between models in future scenarios with significant climate change

have a substantial impact on regional $O_3$ concentration predictions.

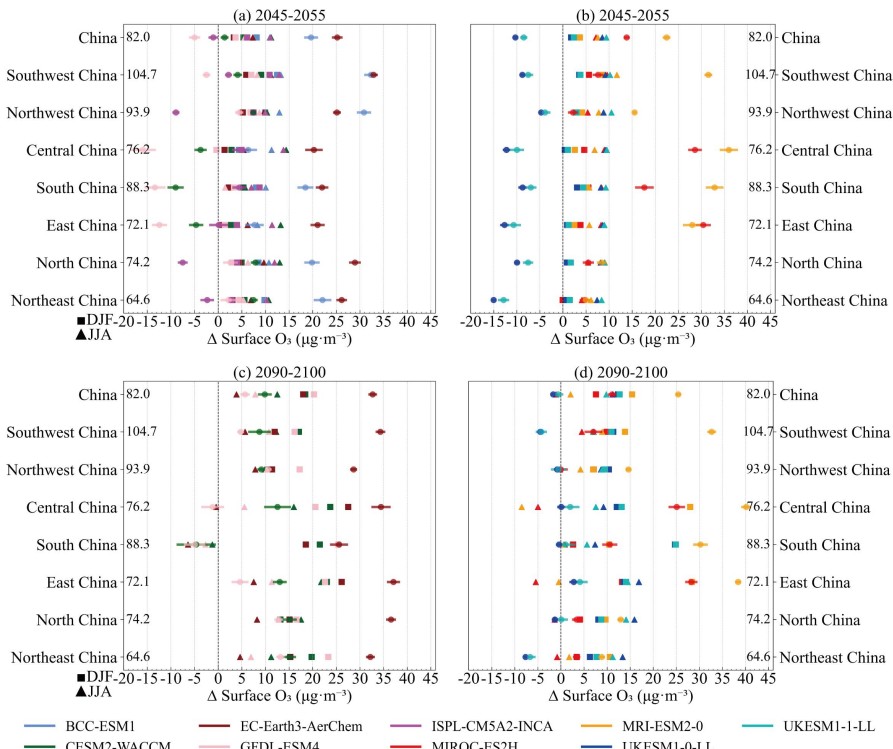

**Figure 13. Changes in the annual and seasonal mean surface O₃ in China and its sub-regions, relative to the 2014–2023 mean, for the SSP3-7.0 scenario used in CMIP6. Each coloured circle represents the annual mean response for an individual model in (a) and (b) for 2045–2055, and in (c) and (d) for 2090–2100, with the coloured bars showing the SD across the annual mean. The seasonal mean responses for DJF and JJA, averaged over the relevant 10-year periods, are shown by squares and triangles, respectively. The Multi-model regional mean for the 2014–2023 period is shown on the left of each panel.**

IPSL-CM5A2-INCA (predictions extending only to 2055) under the370 scenario, which projects that by 2050, surface O₃ concentrations in northern China (including Northeast, North, and Northwest China) will be lower than the 2014–2023 mean, with the most significant decrease expected in North China, where O₃ concentrations are projected to drop by approximately 10%. IPSL-CM5A2-INCA (ECS; 3.6 K) is a model with moderate equilibrium climate sensitivity, showing a moderate response to global temperature increases caused by greenhouse gases. The model simulates relatively high BVOCs emissions during the 2014–2023 period (covering a broader range of BVOCs types), with emissions approximately 4–8 times higher than those of other models (Figure S5), and these emissions have shown a consistent upward trend. However, despite the increase in BVOCs emissions, this model



simulates relatively low $NO_x$ concentrations and surface TAS, resulting in a smaller increase in $O_3$

concentrations, and even a decrease in some regions. This indicates that differences in the magnitude of

climate change and $O_3$ precursor ($NO_x$) variations, as well as the different ways these factors are

coupled in different CMIP6 models, lead to significant differences in the response of the models to

BVOCs emissions. Such differences could directly influence future surface $O_3$ changes, particularly in

localized regions.

CESM2-WACCM and GFDL-ESM4 under the SSP3-7.0 scenario, on the other hand, predict that by

2050, surface $O_3$ concentrations in southern China (including East, Central, and South China) will be

lower than the 2014–2023 annual mean, with the most significant decrease observed in South China,

where $O_3$ concentrations are projected to drop by 9% and 13% in the two models, respectively. Firstly,

CESM2-WACCM (ECS; 4.7 K) and GFDL-ESM4 (ECS; 4.4 K) exhibit higher climate sensitivity,

meaning that their projected temperature increase and water vapor increase are more significant.

Higher temperatures and water vapor content facilitate the generation of OH radicals, which in turn

accelerate $O_3$ destruction reactions (Wild et al., 2020). In tropical and subtropical regions, where both

temperature and water vapor are already high, the presence of these factors may further enhance the

consumption of $O_3$ by OH radicals, a phenomenon particularly evident in South China. Secondly, both

models tend to simulate lower BVOCs emissions (Figure S5), with GFDL-ESM4 showing the lowest

and virtually unchanged BVOCs emissions under the SSP3-7.0 scenario, which may reduce the

formation of $O_3$. Furthermore, the pollutant emissions and atmospheric chemical processes in southern

China differ from those in the north. The southern regions are likely more dependent on photochemical

reactions, which are more active under higher temperatures and stronger solar radiation conditions.

Therefore, CESM2-WACCM and GFDL-ESM4 may simulate a greater number of photochemical

reactions, further accelerating $O_3$ decomposition.

Under the SSP3-7.0 scenario, both UKESM1-0-LL and UKESM1-1-LL project that surface $O_3$

concentrations across China and its sub-regions will decrease by 2050 relative to the 2014–2023 annual

mean, with the most significant decrease observed in Northeast China, where the reduction is projected

to be 23% and 20% for the two models, respectively. Although a moderate increase is projected by

2095, $O_3$ concentrations remain below the 2014–2023 baseline in most regions, except East China.

Compared to other CMIP6 models, UKESM1-0-LL (ECS; 5.4 K) and UKESM1-1-LL (ECS; 4.2 K)

exhibit higher climate sensitivities. The elevated temperatures (Figure S7) and altered climatic



conditions in these models likely enhance $O_3$ degradation, contributing to the generally lower $O_3$

concentrations in their simulations. Moreover, UKESM1-0-LL simulates higher atmospheric $NO_x$

levels, promoting $NO_x$ titration that suppresses $O_3$ formation. This model also projects higher

NMVOCs emissions (Figure S6). Under high-$NO_x$ conditions, the interplay between $NO_x$ and

NMVOCs can modify photochemical $O_3$ production pathways, further inhibiting net $O_3$ formation

(Jiménez & Baldasano; Xing et al., 2011). To further investigate this mechanism, we constructed a

two-dimensional framework based on ground observations, analyzing $O_3$ distribution across China

within the $NO_2$-NMVOCs space (Figure 15). Results indicate that the highest $O_3$ levels occur under

moderate $NO_2$ (20–40 $\mu g \cdot m^{-3}$) and elevated NMVOCs (>500 $t \cdot month^{-1}$) conditions, reflecting a typical

VOC-limited $O_3$ formation regime. In such environments, abundant NMVOCs coupled with relatively

low $NO_x$ levels promote efficient photochemical $O_3$ production. However, when $NO_2$ concentrations

reach 60–120 $\mu g \cdot m^{-3}$, $O_3$ decreases significantly even at intermediate-to-high NMVOCs, indicating

strong inhibition of $O_3$ formation by excess $NO_x$.

The projections of annual mean surface $O_3$ concentrations for China and its sub-regions under the

SSP3-7.0 scenario by BCC-ESM1 (ECS; 4.0K, predictions extending only to 2055),

EC-Earth3-AerChem (ECS; 3.0K), MIROC-ES2H (ECS; 3.6K), and MRI-ESM2-0 (ECS; 5.4K) show

significant consistency, with $O_3$ concentrations in 2050 and 2095 both being higher than the 2014–2023

mean. Among these models, MIROC-ES2H and MRI-ESM2-0 exhibit higher climate sensitivity,

although their projected $NO_x$ and $CH_4$ concentrations are relatively low (Figure S3–S4), they still tend

to predict larger increases in $O_3$ compared to the other models, with the most significant $O_3$ increase

observed in southern China (including East China, Central China, and South China). In contrast,

BCC-ESM1 and EC-Earth3-AerChem predict more noticeable increases in $O_3$ concentrations in

northern China (including Northeast, North, Northwest, and Southwest China). EC-Earth3-AerChem,

with a relatively low ECS, simulates lower $NO_x$ concentrations in China and its sub-regions, while its

$CH_4$ concentrations are the highest among the models (Figure S4). Under low $NO_x$ conditions and a

weak $NO_x$ titration effect, an increase in $CH_4$ promotes $O_3$ formation, leading to $O_3$ accumulation and a

more pronounced increase in simulated $O_3$ concentrations in the model. BCC-ESM1, on the other hand,

tends to simulate higher $NO_x$ concentrations and lower TAS (Figure S3 and S7), which contributes to

the more noticeable $O_3$ increase predicted by this model.



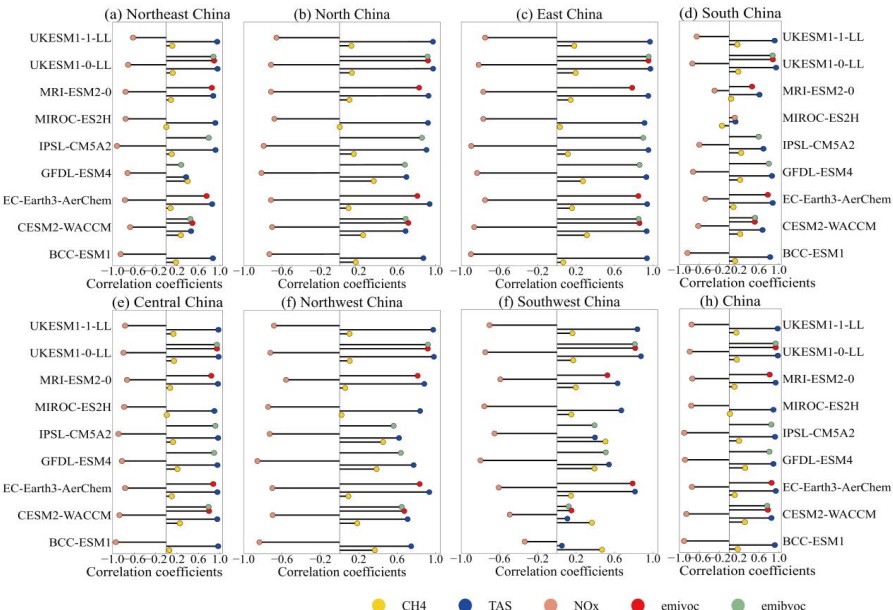

**Figure 14.** Correlation coefficients calculated when comparing future annual mean surface $O_3$ concentrations against individual variables of surface $CH_4$ concentrations, temperature at Surface (TAS), emissions of NMVOCs and BVOCs, $NO_x$ ($NO + NO_2$) concentrations and from individual CMIP6 models over the period 2015 to 2100 in the SSP3-7.0 scenario.

The seasonal responses of different models under the SSP3-7.0 scenario also show variations across sub-regions. Most models predict that surface $O_3$ concentrations increase more in JJA than in DJF. However, some regions exhibit a decreasing trend in $O_3$ concentrations during JJA, which aligns with the findings of Turnock et al. (2020).

As shown in Figure 14, under the SSP3-7.0 scenario, a negative correlation between surface $O_3$ and $NO_x$ concentrations is observed across China and its sub-regions in all the CMIP6 models compared, which may be related to the higher $NO_x$ emissions in the region, leading to the occurrence of $NO_x$ titration. In this scenario, $NO_x$ reacts with surface $O_3$, depleting a significant amount of $O_3$. As $NO_x$ emissions decrease under this scenario (after 2030), the $NO_x$ titration effect weakens, thereby promoting $O_3$ formation at the surface. Consequently, under the SSP3-7.0 scenario, as $NO_x$ emissions decrease, surface $O_3$ concentrations in China and its sub-regions show an increasing trend. Furthermore, most CMIP6 models exhibit positive correlations between other variables (TAS, $CH_4$, NMVOCs, and



BVOCs), indicating that climate change and anthropogenic activities are also important drivers of the

increase in surface $O_3$ concentrations in the region.

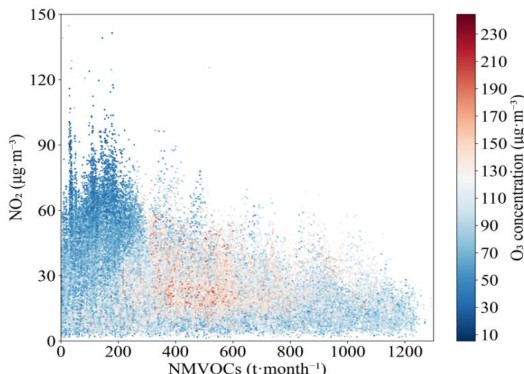


**Fig. 15 Distribution of $O_3$ concentration in $NO_2$-NMVOCs coordinate in China**

The differences in simulations between different CMIP6 models highlight the importance of further

understanding how future $O_3$ concentrations will be influenced by the combined effects of pollutant

emissions (especially with regard to the differences in how $O_3$ precursors, $PM_{2.5}$, and other factors are

coupled and how chemical processes respond across models) and climate change. For example, in the

Central China region, the prediction differences between models are highly significant, with some

models predicting O3 concentrations that could be twice as high as those of others. This discrepancy

reflects the need for future research to focus more on model uncertainty in order to improve the

accuracy of future air quality predictions.

**6 Summary**

In recent years, with the increase in industrial activities and transportation, $O_3$ concentrations in the

atmosphere have risen significantly, leading to profound impacts on the global climate system and

human health. Therefore, an in-depth study of $O_3$ changes over historical periods and under different

future scenarios, as well as its interactions with climate mitigation measures, is essential for assessing

the potential impacts of $O_3$ on the climate and human health. CMIP6 provides a valuable opportunity to

evaluate the simulations of historical and future air pollutant changes using the latest generation of

earth system and climate models, based on the most recent socio-economic development scenarios.



This study, based on the CMIP6 multi-model $O_3$ products and the TAP dataset, analyzes the surface $O_3$

distribution in China and its sub-regions by nine CMIP6 models, as well as the reasons for the

uncertainties in these simulations. It also presents the historical changes of surface $O_3$ in China and its

sub-regions from 1850 to 2014, and evaluates the future changes in surface $O_3$ under different

scenarios (from weak to strong air pollutants and climate mitigation) simulated by the CMIP6 models

for China and its sub-regions. The main conclusions are as follows:

(1)  The MME of CMIP6 simulated $O_3$ concentrations in China are higher in JJA, with an average

value of 105 μg·m$^{-3}$, and lowest in DJF, with a value of 55 μg·m$^{-3}$. Among the seven sub-regions, the

highest $O_3$ concentrations in JJA are found in Central China, while in the other three seasons, the

Southwest region has noticeably higher $O_3$ concentrations compared to other regions, especially in

Tibet. Nine CMIP6 models show significant underestimation in most regions of China, with the most

pronounced underestimation occurring in East China, where the difference is greatest. The Southwest

China is slightly overestimated, while the simulated values for the Northwest China are closest to the

TAP data. All CMIP6 models perform better in JJA simulations, while the differences are larger in DJF.

Among these, EC-Earth3-AerChem, MIROC-ES2H, and BCC-ESM1 produce better simulations, while

UKESM1-0-LL and UKESM1-1-LL show larger discrepancies, with more severe underestimation.

(2)  The Bias, MAE, and RMSE of $O_3$ concentrations simulated by MME on natural land surfaces are

all lower than those on anthropogenic land surfaces. Among these, the simulation performs best on

forest and desert surfaces, while its performance is relatively poorer on urban surface. The Bias of

annual mean surface $O_3$ simulated by MME is lowest under cloudy conditions and highest under partly

cloudy conditions, while for seasonal averages, the variability is smallest under clear-sky conditions in

the summer and largest under less cloudy conditions in the summer. The MME simulations generally

show a decrease in surface $O_3$ concentrations with increasing $PM_{2.5}$ levels (except in JJA), however,

during the DJF, when $PM_{2.5}$ concentrations are high, $O_3$ concentrations increase instead when $PM_{2.5}$

concentrations exceed a certain threshold. Furthermore, The Bias of $O_3$ concentrations simulated by

MME generally increases with the increase in $PM_{2.5}$ concentrations, but once the $PM_{2.5}$ concentrations

exceed a certain threshold value, the Bias then begins to decrease. This also indicates that the effects of

meteorological conditions, subsurface type, cloud cover, pollutant concentration, etc. need to be further

considered when modelling $O_3$ concentration.





(3) Over the entire historical period (1850–2014), the MME-simulated an increase of 39.3 μg·m⁻³ in the annual mean surface $O_3$ concentration in China, with the maximum change of 57.9 μg·m⁻³ (from the MIROC-ES2H model) and the minimum change of 23.1 μg·m⁻³ (from the UKESM1-0-LL model). Before 1950, the annual mean increase in $O_3$ concentration was relatively slow, at only 0.12 μg·m⁻³; however, after 1950, the rate of increase accelerated significantly, with an annual mean increase of 0.28 μg·m⁻³. There are significant discrepancies in the historical changes simulated by different models across various sub-regions, with UKESM1-0-LL tending to simulate the smallest historical changes (smallest in Northeast China), and MIROC-ES2H simulating the largest changes (largest in Central China). Central China is also the region with the greatest diversity of simulated historical changes in surface $O_3$, with a maximum difference of up to 29.6 μg·m⁻³ among multi-model simulations.

(4) Under the weak mitigation scenarios (SSP3-7.0 and SSP5-8.5), the MME projects an increase in surface $O_3$ concentrations across most sub-regions of China under SSP3-7.0,driven by the combined effects of increased air pollutant emissions, higher global $CH_4$ abundance, and climate change. This increase is particularly pronounced in East China, where surface $O_3$ is projected to rise by 19.3 μg·m⁻³ by 2100, a 26.9% increase. Under SSP5-8.5, surface $O_3$ is expected to increase by 2050, but decrease by 2100, especially in South China with a decrease of 12.5 μg·m⁻³ by 2100, a decrease of 14.6%. The SSP3-7.0-lowNTCF predicts relatively small changes in surface $O_3$ across China, with a slight increase (a 4.9 μg·m⁻³ rise by 2100, a 6% increase). In contrast, under the strong climate mitigation and significant air pollutant emission reduction scenario (SSP1-2.6), surface $O_3$ concentrations are projected to decrease across China, particularly in Southwest and South China, where reductions exceed 30 μg·m⁻³. In medium climate mitigation scenarios (SSP2-4.5, SSP4-3.4, SSP4-6.0, and SSP5-3.4-over), surface $O_3$ is expected to increase by 2050, but decrease by 2100. Although SSP1-1.9 represents a weak climate mitigation and weak air pollution reduction scenario, its simulated results show a significant decrease in surface $O_3$, which may be closely related to the reduction of $PM_{2.5}$ in China and the complex effects of climate change.

(5) The projected surface $O_3$ concentrations over China and its sub-regions vary significantly among different climate models, reflecting discrepancies in how these models handle climatic factors (e.g., TAS), atmospheric circulation processes, key chemical reactions (involving $NO_x$ and $CH_4$), and precursor emissions (such as NMVOCs and BVOCs), these differences contribute to substantial uncertainties in regional $O_3$ simulations, particularly in Central China. Under the SSP3-7.0 scenario,



there is a negative correlation between surface $O_3$ and $NO_x$ concentrations, likely due to the occurrence

of the $NO_x$ titration effect. In addition, surface O3 shows a positive correlation with other variables

(TAS, CH4, NMVOCs, and BVOCs) in most CMIP6 models, indicating that climate change and

human activities are also important drivers of surface $O_3$ increases in this region.

This study analyzes the simulation of surface $O_3$ distributions and the reasons for their biases by

different CMIP6 models under various conditions, including different temperatures, cloud cover,

complex underlying surface, and pollutant concentrations in China and its sub-regions. For the

SSP3-7.0, this paper also discusses the interactions between surface $O_3$ concentrations and chemical

species ($NO_x$ and $CH_4$), climate factors (TAS), and natural precursor emissions (NMVOCs, BVOCs),

and analyzes the reasons behind the differences in $O_3$ simulations among the CMIP6 models. It is

noteworthy that, although the driving factors related to $O_3$ changes are important across all models,

significant differences exist in the coupling methods and chemical process responses of the models in

regions with large changes in pollutant emissions (such as $O_3$ precursors, $PM_{2.5}$, etc.) under future

scenarios with significant climate change. A deeper understanding of the mechanisms behind these

differences is crucial for comprehending future $O_3$ trends, developing effective air quality management

strategies, and improving the ability to predict future regional air quality. Additionally, the

ground-based observation data used in this study are relatively limited, and future research should

incorporate more satellite observation data with high spatial and temporal resolution to enrich related

studies and discussions.

*Data availability.* The CMIP6 data can be accessed and downloaded at

https://aims2.llnl.gov/search/cmip6/ (last access: 8 September 2024). TAP data can be and downloaded

at http://tapdata.org.cn/ (last access: 22 July 2024).

*Author contributions.* Shuai Li, Hua Zhang, Qi Chen, Yonghang Chen, Zhili Wang, Qi An and Xinping

Wu designed the study. Xinping Wu and Qi An carried out the data collection. Shuai Li, Qi Chen and

Yonghang Chen carried out the data processing and analysis. Shuai Li, Hua Zhang, and Zhili Wang

assisted with the interpretation of results. All co-authors contributed to writing and reviewing the

paper.



*Competing interests.* The authors declare that they have no conflict of interest.

*Acknowledgements.* We are grateful to the the Earth System Grid Federationr for CMIP6 data support. We further acknowledge the Tsinghua University for TAP data support. We thank the editors and anonymous reviewers, whose comments and suggestions improved the utility and readability of this paper.

*Financial support.* This work was financially supported by the National Key R&D Program of China (grant no. 2022YFC3701202&2017YFA0603502), the National Natural Science Foundation of China (grant no. 42275039), and the S&T Development Fund of CA MS (fund no. 2024KJ021).

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
