# Peer review of "Historical and future changes and present-day uncertainties of ozone in China from CMIP6 models"

_EGUsphere, 2025_

## Referee Comment (RC1)

**Reviewer comments on Li et al., "Historical and future changes and present-day uncertainties of ozone in China from CMIP6 models"**

The paper by Li et al., uses output from multiple CMIP6 models to examine the performance and projection of surface ozone across China. The paper first provides a detailed present-day evaluation of CMIP6 models over China using observational data-fusion product TAP, showing how CMIP6 models tend to underestimate surface ozone over Eastern China but overestimate in the southwest. An analysis is then also performed to look at the influence of cloud, vegetation types and aerosols on the simulation of surface ozone over China. The historical changes in surface CMIP6 models is then discussed showing large increase across China since 1850. Finally, an analysis of future changes in surface ozone across China is presented showing the impact of high mitigation scenarios reduces surface ozone compared to small increases under weak mitigation scenarios.

The contains some useful evaluation of present-day surface ozone concentrations across China and also highlights the future changes in different scenarios. I think the manuscript could be published once the comments below have been addressed.

**Major Comments on sections**

- Abstract At the moment the abstract just reads like a list of results, which is just a
  slightly shorter version of the conclusion. It would be better if the abstract was set out to
  following the traditional format of including a bit on the background/introduction,
  methods, results, and conclusions of the study. The abstract does not currently do this
  and so is hard to get a good summary of the study presented here.
- Methods I think more detail should be included in the methods to help the reader understand how the study was performed.
  - Can you provide a list of which CMIP6 variables that are used in the study and any data references, perhaps in a supplementary table.
  - o Is it worth including any reference to data from the Tier2 scenarios due to the limited available data and also the problem of comparing scenarios when the models providing results are not consistent? Unless there is one model that provides data for all and there is a useful comparison here?
  - Could the results from TAP be included in a first section of the results along with a direct comparison to the CMIP6 results?
  - An ensemble mean for each model is first performed before using to compare to other models. What is variability in surface ozone across ensemble members for a particular model like?
  - The CMIP6 model output (>100 km) is linearly interpolated to that of the TAP data (0.1°). How does the linear interpolation of coarse model data impact on the concentrations produced? Also is there a limitation of using global models at >100km resolution to represent pollutant conditions over highly urbanised regions?
  - O How is the present day uncertainty analysis performed? This seems quite vague and where have the information on temperature, clouds and land surface been obtained from? Is this from the models themselves (which would add additional uncertainty) or somewhere else? Also how does the interpolation of global model data impact this comparison on land surface types?

 Also you mention in section 3.1 that the MME for CMIP6 models is calculated over the period 2014-2023, which CMIP6 experiments has data been obtained from as this period spans both the historical (up to 2015) and scenario (from 2015 to 2100) experiments. More information needed on this.

**• Results -**

- Including a table of statistics to include bias, correlations, trends etc would help improve visibility of results for both the present-day evaluation
- There is a lot of detail on the land surface section about each land cover type, which the global models might have large uncertainties with their process representation. Therefore, I am not sure I get the point of the land cover section other than to highlight models are different from observations due to emissions/deposition processes, especially with the uncertainties in the ability of models to represent these processes. Is there anything specific to East Asia?
- What is the source of the cloud cover data and are there uncertainties in this?
   Can you separate out any analysis into clear sky and cloudy sky?
- Section 3.4 it might be good to mention the main mechanisms that aerosols can impact ozone at the start of the section
- Section 4 Are the discrepancies in the historical trends driven by the 1850 values in each model, as there is larger uncertainty here and also a larger spread in model sensitivities?
- Section 5 Could you compare some of your changes in the scenarios to any other similar studies, perhaps over wider regions? Be careful with the description of the SSP scenarios and what they incorporate. Is it better to just focus on results from the main scenarios with the most data e.g. SSP 126, 245, 370, lowNTCF and 585 due to the data availability in the other scenarios? Can you better link the explanations in the model variabilities to the correlation plot on Figure 14?
- Conclusions A similar issue to the abstract, which is reads like a list of results. It could be made more concise provide a more integrated discussion of the results and what this means in terms of performance and projection. Can you provide more of a discussion of how to take biases learned in present day evaluation to historical and future projections?
- Can the authors comments on the use of a multi-model mean and if any weight should be given to contributions from individual models? For example, UKESM1-0-LL and UKESM1-1-LL basically use the same the chemistry scheme so perhaps their results are giving too much weight to the MME. Also is there an issue in comparing results from experiments with a different number and type of model contributing to the MME (e.g. 9 models for SSP370 and 5 models for SSP126 or see line 535)?

**Minor Comments on Structure, Figures and Tables**

Suggestion of slightly re-wording title to:

"Historical and Future changes of surface ozone over China from CMIP6 models, including an assessment of present-day uncertainties in model prediction."

Should section 3.1 just be labelled as a present-day model evaluation section? I am not sure where temperature fits in (apart from the seasonal cycle)

Section 5 is very long, can this be split up in to different sub-sections to make the article read better?

Table 1 – I think the UKESM1-1-LL reference should be

https://gmd.copernicus.org/articles/16/1569/2023/ . Also could you include a few details on the models such as resolution here?

Figure 1 – It is very hard to see the aerosol components on this chart so I would just not show them especially as this study is meant to focus on ozone. Also the number are also hard to read. Perhaps the statistics could be included in a table somewhere else? I wonder if it is worth including the CMIP6 multi-model mean for a direct comparison of temporal changes?

Figure 2 – I might to suggest to use different colour bars for each column of plots that show different metrics as having the same colour scale for all is currently a bit confusing

Figure 3 – perhaps the MME could be highlighted more for a direct comparison to TAP, with the individual models made to be more transparent or smaller.

Figure 7 – More details in the caption on the Figure, what is (b) showing? and very similar colours to those on Figure(a) so confusing

Figure 8 – More details in the caption on the Figure

Figure 10 – similar to another Figure, could the MME be made more prominent?

Figure 11 – Why do some of the scenarios start about 30  $\mu$ g m-3 above the zero line?

Figure 15 – Could you make better use of this of this Figure to help explain where future changes sit in terms of NOx/VOC ratio?

**Minor Comments on Text**

Line 16 - replace "current" with "present-day"

Line 35 – insert "gas" between "trace" and "components"

Line 35-39 – Sentence is quite long. Could be broken up into two parts to read better

Line 50 – I am not sure "organisms" is the right word here. Maybe replace with "ecosystems"?

Line 51 – "past few years" – can you be more specific about the time period?

Line 76 -77 – Are there are more recent observations to show the continual decline in Air Quality due to increasing ozone? What time period are you referring to in this sentence?

Line 80 – What is the "Dual-Carbon" strategy and how does this relate to air pollutants since it seems to be referring to carbon?

Line 92 – What about other models such as Chemical Transport Models (CTMs), are these not also used as well?

Line 101 – Linked to the above, perhaps at this stage it might be worth mentioning other multi-model initiatives that have attempted to understand surface ozone such as HTAP (Hemispheric Transport of Air Pollutants) and its regional counter-part MICS-Asia

(https://acp.copernicus.org/articles/special\_issue390.html). Additionally some comments on the TOAR (Tropospheric Ozone Assessment Report) multi-model comparison would be useful.

Line 113 - Include "in the present-day." After "current uncertainties"

Line 171 – Are these average values for the full 24 years of the TAP data? If so is this useful given the large changes in both  $O_3$  and  $PM_{2.5}$  that have occurred over this period?

Line 198 – also the largest standard deviation as well

Line 213 – So does this imply that the TAP data has been made with a stronger southeast Monsoon? Where has the meteorological data come from?

Line 217 – What about source of ozone from the free troposphere and stratosphere that could influence this region?

Line 221 - Why is ozone always so high is Northwest China along with no seasonal cycle? Is this because of lower local emissions and higher import from extra-regional sources?

Line 223 – Similarly why is ozone much lower in north east China?

Line 229-230 – I would say that the SD is only largest in DJF in in Sichuan basin. I would say there is larger SD across more regions in JJA.

Line 235 – Potentially larger biases in MAM and SON?

Line 237 – It maybe worth reporting all the statistics (not just the bias ones here) in a table for easy reference to the reading. Same with the correlations on Line 246

Line 249 – I think delayed maximum ozone is true in some regions (North China) but perhaps more similar in others (Northwest China)

Line 250-252 – It is interesting that an underestimation of ozone concentrations is reported here whereas, in Figure 2 it seems like outside of Eastern China there is a lot of overestimation of ozone. Perhaps the MME could be made clearly on Figure 3 so that the under/over estimation across regions is clearer.

Line 252 – the biases in UKESM1 are explored further in this paper which might help explain things <a href="https://acp.copernicus.org/articles/22/12543/2022/">https://acp.copernicus.org/articles/22/12543/2022/</a>

Line 289 – Are the biases in the MME for different seasons shown? If not could they be included on Figure 5?

Line 325 – Do the natural land surface have consistent physical and chemical properties with less human influence? What about deforestation on forests and BVOCs/Fires which can mean quite complicated processes.

Line 329 - What about the impact of deposition on land surfaces here?

Line 335-339 – If the models do not account for accurate deposition on snow/ice is it worth this much detail and is this a big issue over China?

Line 364-365 – Is wet deposition a large removal term for ozone?

Line 366 – Which precursors?

Line 380-384 – Isn't this what the models try to do? Can you be a bit more specific as this statement is very general.

Line 390-393 - Can you be more specific about how aerosols alter the chemical composition?

Line 401 – so this is suggesting that the aerosol impact on ozone formation is smaller in summer/autumn?

Line 406-407 – which pathways?

Line 464 – "driven solely by precursor emission variations" – I am not sure I understand this point

Line 471-473 – yes this might be an impact but can you say what time period this might be more important over?

Line 482 – could also present percentage changes here too?

Line 492 – Why does SSP245 increase by 2055?

Line 493 – Is the exceedance of 10  $\mu$ g m-3 already mentioned earlier?

Line 505-506 – Is there not also large differences over South China?

Line 507-10 - SSP3-7.0-lowNTCF scenario only targets air pollutant controls and nothing to do with carbon controls. Does the lowNTCF scenario not actually increase the contribution to climate change <a href="https://acp.copernicus.org/articles/20/9641/2020/">https://acp.copernicus.org/articles/20/9641/2020/</a>?

Line 525-526 – SSP585 actually has strong air pollutant controls https://doi.org/10.1016/j.gloenvcha.2016.05.012

Line 530-531 – do the SSP585 and SSP370-lowNTCF have similar air pollutant emissions and climate change?

Line 539-545 – SSP1-1.9 has strong controls on climate and air pollutants. Careful with the description of effects from this scenario

Line 549-550 – SSP5-3.4-over – this is an overshoot scenarios so will the time evolution be particularly different?

Line 592-593 – why are NOx emissions so different between the models?

Line 595 – Similar to above why are CH4 concentrations so different given that the input data should be the same?

Line 658 – Is this still going to be a VOC-limited regime in the future?

Line 660 - Is this still what is happening in the future? Where does the future NOx and VOCs fit on this diagram?

Line 685-688